# Intrasexual Competition in Women's Likelihood of Self-Enhancement and Perceptions of Breast Morphology: A Hispanic Sample

Ray Garza [1,*] and Farid Pazhoohi [2]

1    Department of Psychology and Communication, Texas A&M International University, Laredo, TX 78041, USA
2    Department of Psychology, University of British Columbia, Vancouver, BC V6T 1Z4, Canada
*    Correspondence: ray.garza@tamiu.edu

**Abstract:** Women's breasts are considered sexually attractive because they may infer a woman's residual reproductive value. Given that men find women's breasts attractive, women may compete with other women to enhance their physical attractiveness when primed with an intrasexual competitive cue. The current study investigated women's intrasexual competition when viewing variations in breast morphology. Women (*N* = 189) were randomly assigned to a partner threat condition and shown images of women's breasts that included variations in breast size, ptosis (i.e., sagginess), and intermammary distance (i.e., cleavage). Women were more likely to report an increase in enhancing their appearance, wearing revealing clothing, dieting and exercising, and perceiving the breasts as a sexual threat as a function of larger breast sizes with low ptosis and intermediate distances. The partner threat prime did not play a role in ratings. Interestingly, there was a moderating role for women's dispositional levels in intrasexual competition. Women with higher levels of intrasexual competition were more likely to enhance their appearance when viewing large breast sizes. The study points to the role that breast morphology indicative of residual reproductive value has on increasing enhancement strategies.

**Keywords:** intrasexual competition; breast size; breast ptosis; intermammary distance; enhancing appearance

## 1. Introduction

According to sexual selection theory, traits that are preferred in the opposite sex influence intrasexual competition in the same sex [1]. Female breasts are sexually dimorphic traits that are thought to be sexually selected, as men find them attractive, sexually arousing, and show variation in their preferences for breasts that are cues to youthfulness and reproductive value [2]. Breast morphology is also thought to play a role in the perception of women's residual reproductive value, as men and women have shown to associate specific features of women's breasts as being associated with fecundity [3,4]. Nonetheless, given that female breasts are associated with intersexual selection [5], there is also evidence to suggest that breast morphology is instrumental in intrasexual competitive displays in women, as women may compete with other women to appear more desirable or prevent their partners from desertion. The current study focuses on women's intrasexual competitive displays by focusing on breast morphology, such as breast size, ptosis (i.e., sagginess), and intermammary distances (i.e., cleavage).

Research on women's breast morphology has primarily focused on men's preferences for women's breasts. According to the nubility hypothesis, breasts serve as an indicator of sexual maturity, as women's breast sizes change from pre- to post-pubescence [2], suggesting that breast morphology can be a reliable cue to age and reproductive value. Eye-tracking research has shown that breast morphology captures men's visual attention compared to other body regions [6–8]. Compared to smaller breasts, men have shown preferences for

larger breast sizes [4,9,10], perhaps because larger breasts may be associated with fertility and reproductive success [4]. However, men have shown variation in their preferences for women's breasts, such as finding women with small [11,12], medium [13–15], and large breasts [4,10,16,17] physically attractive. It has also been suggested that preferences for breast sizes is a Western preference, as men's preferences for large breasts were not a universal preference among men from different cultural backgrounds [16,18,19]. It is noteworthy to mention that there is evidence that obesity, large, and very large breast sizes appear to hinder the ability to lactate and breastfeed [20–23].

Furthermore, other breast features, such as ptosis (i.e., sagginess) and intermammary distance (i.e., cleavage) also influence men's preferences for women's breasts. Breast ptosis has been shown to be associated with men's perceptions of women's attractiveness, fecundity, health, and youthfulness. Women with non-ptotic breasts (i.e., firm and upright) are perceived to be more attractive [14,24–26] compared to high ptotic (i.e., saggy) breasts. This may be due to the perception that ptosis has on women's age, as non-ptotic breasts are perceived to be younger while ptotic breasts are perceived to be older. Ptotic breasts are also associated with a woman having multiple pregnancies [23] and being in poorer health [27,28]. Women with a smaller intermammary distance (i.e., cleavage) have been linked to increased levels of attractiveness, health, and perceived to be older [3,10].

It has been suggested that women have evolved psychological mechanisms to be able to detect and compete with other women [29]. Given that men prioritize physical features in mate choice, women may compete with members of the same sex, such as enhancing their appearance or displaying features that correspond to men's evolved mate preferences [30–32]. They may exploit men's preferences for physical attractiveness when in direct competition with other women to increase their likelihood in mating success. For instance, women have shown to engage in competitive tactics when competing with other women who are physically attractive or show cues of sexual receptivity or interest. They derogate women who dress provocatively [33] or show cues of sexual interest, such as wearing a red dress [34]. Women are also likely to rate women negatively on traits such as intelligence, friendliness, and attractiveness, if they consider rivals as a threat to their current relationship [35]. Women employ different behavioral strategies to indicate their sexual interests to men [36,37]. For example, recent research has for the first time proposed and shown that women who display increased lumbar curvature and arch their backs, i.e., lordosis behavior, are perceived to be sexually receptive and more attractive [38,39]. Women consider women in lordosis posture threatening to their current relationship, which is positively influenced by perceivers' own dispositional levels of intrasexual competitiveness [39]. This latter points to the function of wearing high heels in women as a strategy for competition with other women through enhancement of physical attractiveness and signal of sexual receptivity [40–43].

Self-enhancement is a common stagey used in intrasexual competition in women [44]. Women who report a higher level of intrasexual competition report a higher frequency of make-up usage and spend more money per month on make-up and beautification products [45]. When primed about competing for a romantic partner and attractive member of the same sex, women show an increase likelihood to take diet pills [32]. Social comparison to others has also shown to increase the likelihood of taking diet pills, albeit mediated by envy [46]. Taken together, enhancement strategies have been shown to be influenced by competitive cues, such as intrasexual competitive priming.

In relation to breast morphology, women consider large breasts to be more feminine, attractive, and threatening [47], suggesting that they are attentive to traits that men desire in the opposite sex. Indeed, women are more likely to engage in risky surgical procedures [48], and hold more favorable attitudes towards enhancement procedures [49]. Since antiquity some women have engaged in nonsurgical cosmetic approaches in breasts size and intermammary distance enhancements through wearing corsets and pushup bras [50,51]. Breast augmentation continues to be the most popular cosmetic procedure, specifically breast size and breast lift augmentation [48]. Since men show a preference for larger breasts

that are non-ptotic, women may enhance these traits to compete with other women and increase their attractiveness and desirability in mating [52,53]. In one study examining the link between women's breast size, ptosis, intermammary distance, and their ratings of being a threat and introducing a woman who possesses these features to their current partner, women with large non-ptotic breasts were more likely to be rated as a threat and less likely to be introduced to a woman's current partner [4]. This was strengthened in the experimental condition, where women who read a prompt on partner threat (i.e., imagining a scenario where another attractive female is interested in the woman's partner) were less likely to introduce their current partner to a woman with attractive breasts. Nonetheless, this suggests that women with attractive breast features are potentially considered as threats and may attempt to derogate or shield their current partners. Whether women also attempt to enhance their appearance in the presence of women with variations in breast morphology has not been explored.

The aim of the current study was to investigate women's ratings on intrasexual competition traits when viewing women's breast morphology with variations in breast size, ptosis, and intermammary distance. Importantly, we focused on an underrepresented population in the evolutionary psychology literature, which are Hispanic women. While there are widespread variations in different domains of human psychology across cultures and populations [54], Latin American and Caribbean samples represent about 2.5% in evolutionary psychology [55]. Using a sample of Hispanic participants, we also aimed to provide more diversity to this literature in understanding human preference from an evolutionary perspective. We also examined if ratings would differ when exposed to an experimental condition of partner threat, as a partner threat condition may amplify and make competition with other women more salient. The study also explored the role of women's individual differences in intrasexual competition when making ratings. The study hypothesized that women would report higher ratings of appearance enhancement, wearing revealing clothing, dieting and exercise, and sexual threat. It was also hypothesized that those ratings would be stronger in the experimental (partner threat) condition compared to the control condition. Lastly, we hypothesized that individual differences in intrasexual competition, as measured by the Intrasexual Competition Scale [56], would positively correlate with appearance enhancement, wearing clothing, dieting and exercise, and sexual threat.

## 2. Materials and Methods

### 2.1. Participants

An a priori power analysis using G*Power (Version 3.1) ($f$ = 0.15, $a$ = 0.05) revealed that a sample size of 182 participants would be an adequate sample to detect a small to medium effect size. A total of 189 ($M_{age}$ = 23.46, $SD_{age}$ = 5.16) self-identified heterosexual women participated in the online experiment in exchange for course credit. The sample demographics were predominantly Hispanic ($n$ = 178), White ($n$ = 9), African-American ($n$ = 1), and Asian-American ($n$ = 1) women. A total of 100 participants reported their relationship status as single, and 89 reported to be in a relationship.

### 2.2. Measures

#### 2.2.1. Partner Threat Prime

The partner threat prime used was from Fischer and Archibald [35], which describes four different scenarios of a woman experiencing another attractive woman interested in her partner. The scenario describes a situation where women are to imagine another attractive woman showing interest in their current partner. The scenarios are depicting competition in different contexts, such as a partner showing interest in another woman's text message, or a woman showing interest in her partner at a party. For the between subject manipulation of partner threat, all four primes were used in one condition to increase the saliency of the intrasexual competition prime, as has been used in previous research [35,49]. The scenarios are shown in Appendix A.

### 2.2.2. Breast Stimuli

The breast stimuli used were from Pazhoohi et al. [10], which are images that show the chest region only, and include 4 breast size levels (A, B, C, D), 3 levels of ptosis (non-ptotic, low ptotic, high ptotic), and 3 levels of intermammary distance (small, intermediate, large), see Figure 1. In total, there were 36 images of breasts.

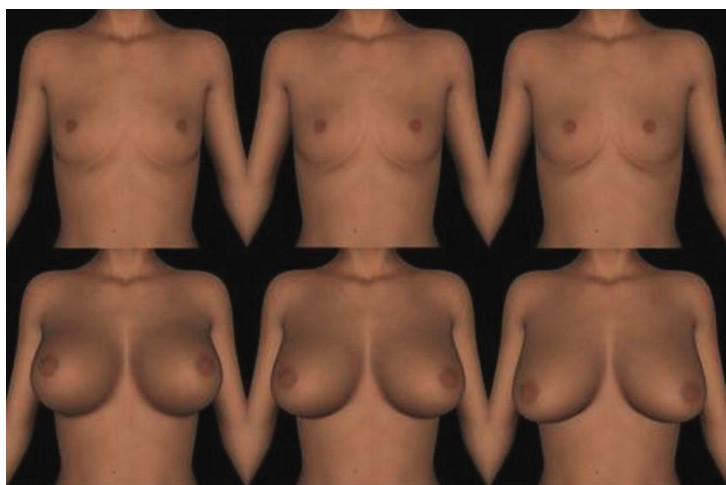

**Figure 1.** Examples of breast stimuli used. The top row depicts stimuli with three levels of intermammary distance for A-cup breasts, and the bottom row depicts three levels of ptosis for D-cup breasts.

### 2.2.3. Intrasexual Competition Scale

The Buunk and Fisher [56] Intrasexual Competition scale (ICS) was used to measure individual differences in women's level of same-sex competitiveness. It is a 12-item instrument measured in a 7-point Likert scale on responses to statements, such as, "I can't stand it when I meet another woman who is more attractive than I am", where higher scores indicate higher intrasexual competitiveness and lower scores indicate lower levels of intrasexual competitiveness. The Cronbach's alpha for the intrasexual competition scale was $\alpha$ = 0.91, which indicated very good reliability.

### 2.2.4. Dependent Variables

There were four dependent variables. Participants were asked to answer their likelihood to the following statements, "likelihood of enhancing appearance", "likelihood of wearing revealing clothing", likelihood of dieting and exercising", and "likelihood that the woman viewed was a sexual threat". The variables were measured on a 1–7 Likert scale, where "1 = not very likely", and "7 = very likely".

### *2.3. Procedure*

Participants signed up for the study announced on the Texas A&M International University's SONA system research management system. The study was announced to women only using a pre-screener, and participants were directed to a Qualtrics link. Upon consent, participants completed a sociodemographic questionnaire, followed by the Intrasexual Competition Scale. When complete, they were randomly assigned to either a control or experimental condition. In the experimental condition, they were presented with four partner threat scenarios used in Fisher and Archibald [35] presented sequentially in separate pages. In the control condition, they were not exposed to any of the partner threat scenarios and were given instructions that they were to view each image carefully and provide ratings for each image. We followed the same approach from previous studies [3,4,10], where they viewed the 36 breast images, and indicated their likelihood to enhance their appearance, wear revealing clothing, diet and exercise, and the likelihood of the woman viewed being a sexual threat. The ratings were presented at the bottom of the same page of the breast image presentation. Once complete with the ratings, they

were asked a question on how realistic the breast images appeared using a 1–7 Likert scale, where "1 = not realistic at all" to "7 = very realistic".

## 3. Results

### 3.1. Manipulation Check

To measure the effectiveness of the partner threat prime, we ran an independent *t*-test for participants responses to feeling threatened when exposed to the partner threat and control condition. Women indicated a higher level of feeling threatened in the partner threat condition ($M = 5.61$, $SD = 1.94$) compared to the control condition ($M = 2.87$, $SD = 1.80$, $p < 0.001$) indicating that the experimental manipulation of partner threat was effective. At the end of the breast stimuli presentation, women answered the following: "On a scale from 1 = not at all realistic to 7 = very realistic, how would you rate the breast images that you viewed?" The overall realism rating for the breasts was $M = 5.03$, $SD = 1.57$.

### 3.2. Data Analysis

For all dependent variables, we ran a 2 (partner threat: threat vs. control) × 4 (breast size: A-, B-, C-, D-cup) × 3 (ptosis: non-ptotic, low ptotic, high ptotic) × 3 (intermammary distance: small, intermediate, large) mixed ANOVA with partner threat as the between subjects factor and breast size, ptosis, and intermammary distance as within subjects factors. To correct for a violation in sphericity, all mixed ANOVAs were run with a Huynh–Feldt correction. We used a corrected alpha level of $p = 0.013$ ($0.05/4 = 0.013$) to adjust for experiment-wise error rate for testing four dependent variables. For pairwise comparisons, a Bonferroni correction was used. For testing the moderating role of individual differences in intrasexual competition on ratings when exposed to female breasts, a linear mixed effect model (LME) was used using maximum likelihood estimation. For the LME models, intrasexual competition, breast size, ptosis, intermammary distance, and the two-way interactions between intrasexual competition and breast size, intrasexual competition and ptosis, and intrasexual competition and intermammary distance were entered as fixed effects. Subject ID (i.e., participants) were entered as a random effect. We report the unstandardized betas for all of the significant main effects and interactions involving the moderator of intrasexual competition.

#### 3.2.1. Enhancing Appearance

There was a significant main effect for breast size, $F(3, 231.05) = 4.59$, $p = 0.003$, $\eta^2_p = 0.02$. Women were more likely to enhance their appearance when viewing D-cup ($M = 4.03$, $SE = 0.13$), compared to C-cup ($M = 3.73$, $SE = 0.11$), B-cup ($M = 3.58$, $SE = 0.10$), and A-cup ($M = 3.54$, $SE = 0.14$) breasts. A significant main effect for intermammary distance, $F(2, 358.58) = 5.51$, $p = 0.004$, $\eta^2_p = 0.02$, showed that women were more likely to enhance their appearance when viewing breasts with large intermammary distances ($M = 3.79$, $SE = 0.08$) compared to small ($M = 3.66$, $SE = 0.08$), but not intermediate intermammary distances ($M = 3.71$, $SE = 0.08$). A marginal significant main effect for ptosis, $F(2, 312.14) = 3.83$, $p = 0.02$, $\eta^2_p = 0.02$, revealed that women were more likely to enhance their appearance when viewing low-ptotic ($M = 3.80$, $SE = 0.09$) compared to non-ptotic ($M = 3.65$, $SE = 0.08$), but they were not significantly different when compared to high-ptotic breasts ($M = 3.71$, $SE = 0.09$). The results were qualified by a ptosis by size interaction, $F(5.21, 970.42) = 2.76$, $p = 0.01$, $\eta^2_p = 0.01$, where women's ratings of enhancing appearance were C-cup breasts with low-ptosis ($M = 3.89$, $SE = 0.12$) compared to C-cup non-ptotic breasts ($M = 3.58$, $SE = 0.13$), but not significantly different than C-cup breasts with high-ptosis ($M = 3.71$, $SE = 0.12$, see Figure 2a. There were no other significant main effects or interactions, nor main effects or interactions with the experimental condition.

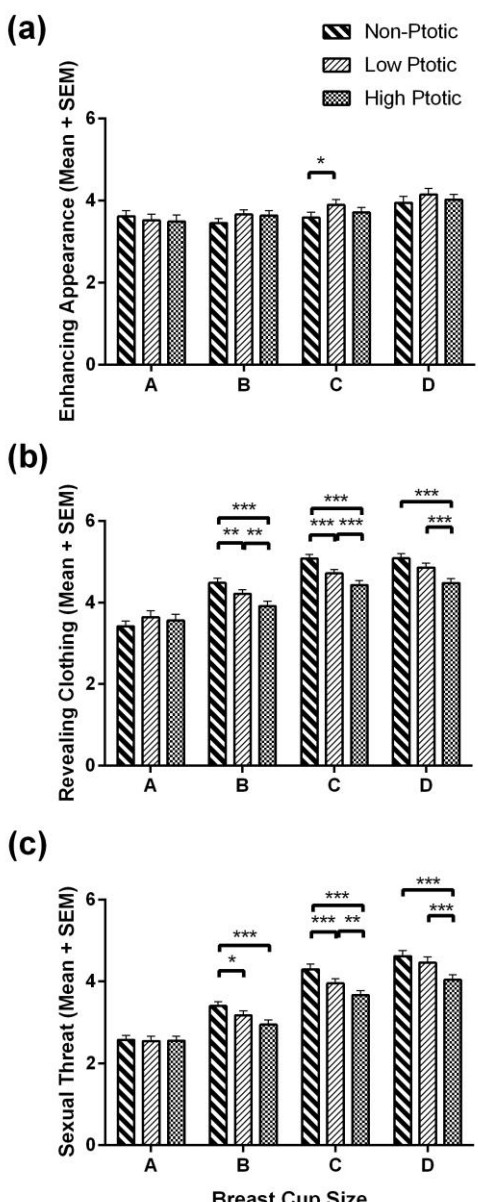

**Figure 2.** Women's likelihood of (**a**) enhancing their appearance, (**b**) wearing revealing clothing, and (**c**) perceiving breasts as sexually threatening across breast sizes (A–D-cup) and levels of ptosis. Note: $p < 0.05$ *, $p < 0.01$ **, $p < 0.001$ ***.

### 3.2.2. Revealing Clothing

There was a significant main effect for breast size, $F(1.26, 232.21) = 47.98$, $p < 0.001$, $\eta^2_p = 0.21$. Women were more likely to wear revealing clothing when viewing women with D-cup ($M = 4.81$, $SE = 0.10$) compared to B-cup ($M = 4.20$, $SE = 0.10$) and A-cup breasts ($M = 3.54$, $SE = 0.14$), but not C-cup breasts ($M = 4.74$, $SE = 0.09$). A significant main effect for intermammary distance, $F(1.89, 349.48) = 11.43$, $p < 0.001$, $\eta^2_p = 0.06$, indicated that women were more likely to wear revealing clothing when viewing women with intermediate ($M = 4.40$, $SE = 0.08$) compared to small intermammary distances ($M = 4.23$, $SE = 0.08$) but not significantly different when compared to large intermammary distances ($M = 4.33$, $SE = 0.08$). A ptosis main effect, $F(1.86, 343.14) = 52.78$, $p < 0.001$, $\eta^2_p = 0.22$, showed that wearing revealing clothing were influenced by viewing non-ptotic breasts ($M = 4.51$, $SE = 0.08$) compared to low- ($M = 4.35$, $SE = 0.08$) and high-ptotic breasts ($M = 4.09$, $SE = 0.09$). There was a significant breast size by intermammary distance, $F(5.64, 1038.74) = 4.20$, $p < 0.001$, $\eta^2_p = 0.03$, and breast size by ptosis interaction,

$F(5.60, 1031.92) = 18.30$, $p < 0.001$, $\eta^2_p = 0.09$. Women were more likely to wear revealing clothing when viewing non-ptotic breasts across breast sizes B–D-cup, with higher ratings reported for C-cup non-ptotic breasts, see Figure 2b. Women were more likely to wear revealing clothing when viewing C-cup breasts with intermediate distance ($M = 4.84$, $SE = 0.09$) compared to C-cup with small distances ($M = 4.62$, $SE = 0.10$) but not C-cup with large distance ($M = 4.73$, $SE = 0.10$), see Figure 3a. There were no other significant main effects or interactions, nor main effects or interactions with the experimental condition.

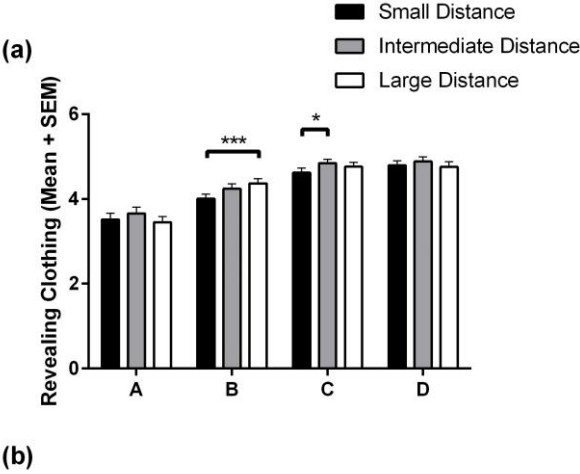

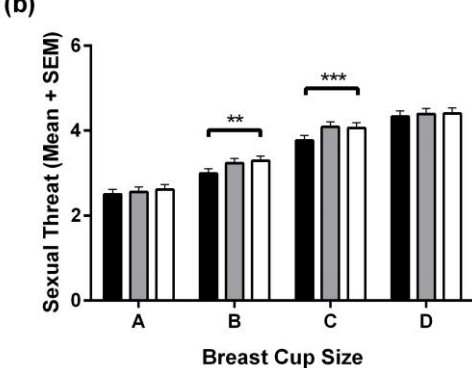

**Figure 3.** Women's likelihood of (**a**)wearing revealing clothing and (**b**) perceiving breasts as sexually threatening across breast sizes (A–D-cup) and intermammary distances. Note: $p < 0.05$ *, $p < 0.01$ **, $p < 0.001$ ***.

3.2.3. Diet and Exercise

For diet and exercise, a main effect for breast size, $F(1.28, 241.07) = 4.01$, $p = 0.03$, $\eta^2_p = 0.02$, and intermammary distance, $F(2, 374) = 3.34$, $p = 0.03$, $\eta^2_p = 0.02$, were trending towards significance at the $p = 0.013$ level. Women reported higher levels of dieting and exercise as breast sizes increased (A-cup: $M = 3.89$, B-cup: $M = 3.86$, C-cup: $M = 3.68$, D-cup: $M = 3.53$), but the differences were not significant from each other. For intermammary distance, women reported higher levels of dieting and exercise for the intermediate and large intermammary distances (Both $M's = 3.77$, $SE = 0.08$), but the differences were not significantly different when compared to the small distances ($M = 3.68$, $SE = 0.08$). The interaction between breast size and intermammary distance, $F(5.54, 1036.81) = 2.28$, $p = 0.03$, $\eta^2_p = 0.01$, were not significant at the $p = 0.013$ level. There were no other significant main effects or interactions, nor main effects or interactions with the experimental condition.

3.2.4. Sexual Threat

A significant main effect for breast size, $F(1.39, 260.37) = 120.01$, $p < 0.001$, $\eta^2_p = 0.39$, revealed that D-cup breasts ($M = 4.37$, $SE = 0.12$) were rated as being sexually threatening compared to C- ($M = 3.97$, $SE = 0.11$), B- ($M = 3.17$, $SE = 0.09$), and A-cup breasts ($M = 2.55$, $SE = 0.11$). There was a significant main effect for intermammary distance,

$F(1.99. 372.39) = 17.08$, $p < 0.001$, $\eta^2_p = 0.08$. Women rated breasts with large intermammary distances ($M = 3.59$, $SE = 0.09$) higher on being a sexual threat compared to small distances ($M = 3.39$, $SE = 0.09$) but not different compared to intermediate distances ($M = 3.56$, $SE = 0.09$). There was a significant main effect for ptosis, $F(1.72, 321.71) = 43.89$, $p < 0.001$, $\eta^2_p = 0.19$. Non-ptotic breasts ($M = 3.72$, $SE = 0.09$) were rated higher on being a sexual threat compared to low-ptotic ($M = 3.53$, $SE = 0.09$) and high-ptotic breasts ($M = 3.30$, $SE = 0.09$). A size by ptosis interaction, $F(5.61, 1049.20) = 8.96$, $p < 0.001$, $\eta^2_p = 0.04$, revealed that non-ptotic D-cup breasts ($M = 4.62$, $SE = 0.14$) were rated higher on being a sexual threat compared to high ptosis ($M = 4.04$, $SE = 0.12$) but not different compared to low ptosis ($M = 4.46$, $SE = 0.13$), see Figure 2c. A marginal significant interaction between breast size and intermammary distance interaction, $F(5.67, 1061.54) = 2.37$, $p = 0.03$, $\eta^2_p = 0.01$, revealed that C-cup breasts with large intermammary distances were rated higher on sexual threat ($M = 4.08$, $SE = 0.11$) compared to C-cup breasts with small distances ($M = 3.76$, $SE = 0.11$), see Figure 3b. There was a significant interaction between partner threat condition and ptosis but post-hoc comparisons did not reveal any significant effects across conditions (see supplemental material). There were no other significant main effects or interactions, nor main effects or interactions with the experimental condition.

### 3.2.5. Individual Differences in Intrasexual Competition

Linear mixed effect models were run to test the moderating role of intrasexual competitiveness in women and their ratings when exposed to women's breasts. In all moderation analyses, the lowest level (i.e., A-cup, non-ptosis, small intermammary distance) of the repeated measures variable was used as the reference category. For appearance enhancement, women's intrasexual competitiveness moderated the likelihood of enhancing appearance when viewing D-cup ($b = 0.05$, $SE = 0.004$, $p < 0.001$), C-cup ($b = 0.03$, $SE = 0.004$, $p < 0.001$), but not B-cup breasts ($b = 0.006$, $SE = 0.004$, $p = 0.12$), see Figure 4. To better understand this interaction, we probed the interaction at $-1SD$, the mean, and $+1SD$ from the mean of intrasexual competition. At lower levels ($-1SD$) of intrasexual competition, women reported a higher likelihood of enhancement appearance when viewing A-cup breasts compared to C- ($b = -0.24$, $SE = 0.08$, $p = 004$) and D-cup breasts ($b = -0.16$, $SE = 0.08$, $p = 0.06$), but not B-cup breasts ($b = -0.05$, $SE = 0.08$, $p = 0.52$). At the mean of intrasexual competition, women were more likely to enhance their appearance when viewing C-cup ($b = 0.19$, $SE = 0.06$, $p = 0.001$) and D-cup breasts ($b = 0.51$, $SE = 0.06$ $p < 0.001$), but not B-cup breasts ($b = 0.03$, $SE = 0.06$, $p = 0.54$) when compared to A-cup breasts. At higher levels of intrasexual competition, ($+1SD$) women reported a higher level of enhancing appearance when viewing C- ($b = 0.64$, $SE = 0.08$, $p < 0.001$) and D-cup breasts ($b = 1.18$, $SE = 0.08$, $p < 0.001$), but not B-cup breasts ($b = 0.13$, $SE = 0.08$, $p = 0.13$). There were no significant interactions for women's likelihood of wearing revealing clothing when exposed to breast morphology. For likelihood of dieting and exercise, there were no significant interactions; however, there was a significant main effect for women's levels of intrasexual competition ($b = 0.01$, $SE = 0.007$, $p = 0.02$). Women who reported higher levels of intrasexual competition were more likely to indicate a higher likelihood to diet and exercise when viewing all the breast images. In ratings of women's breasts being a sexual threat, there was a significant main effect for intrasexual competition ($b = 0.02$, $SE = 0.002$, $p = 0.005$). Women with higher levels of intrasexual competition were more likely to rate woman in the images higher on being a sexual threat. There were no other significant interactions with intrasexual competition and breast morphology.

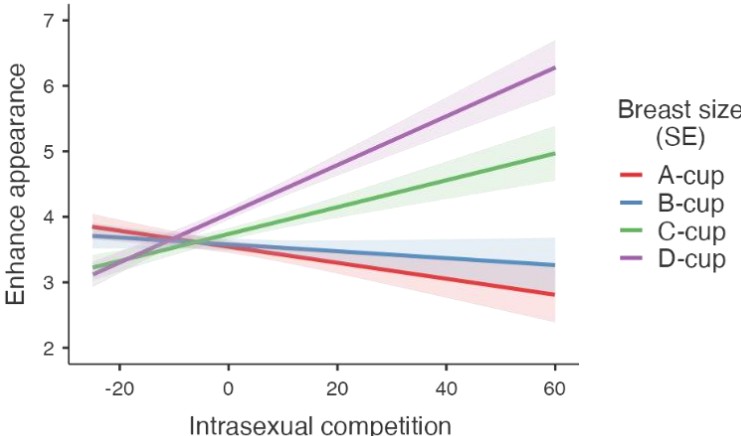

**Figure 4.** Intrasexual competition moderating the relationship between breast sizes and enhancing appearance.

## 4. Discussion

The current study showed that women's intrasexual competition (i.e., enhancing appearance, and perceiving women's features as threatening) were influenced by breast morphology. More specifically, they were affected by breast features that men find attractive and associate with residual reproductive value. Women were more likely to enhance their appearance when viewing low ptotic, C-cup breasts. They were more likely to wear revealing clothing when viewing non-ptotic C- and D-cup breasts, D-cup breasts with intermediate intermammary distances, and report dieting and exercising when viewing D-cup breasts with intermediate distances. Furthermore, women rated C-cup breasts with large distances and C-cup non-ptotic breasts as being sexually threatening. Regarding sexual threat, women may be attentive to these breast features, as men consider non-ptotic, large breasts attractive, and women may guard their current partners from women who possess these physical features. Interestingly, we did not find any evidence that the experimental prime of partner threat intensified women's ratings across the dependent variables, even though women did report the partner threat primes as being threatening compared to the control condition. This may suggest that priming intrasexual competition may not affect ratings, as women may always be attentive to possible threats and competitors in the environment, and this may influence behaviors associated with enhancing their appearance. A partner threat prime was not effective in perceptions of women's ratings of breast morphology in a previous study [4]. Nonetheless, we did find some evidence that women's dispositional intrasexual competition did play a role in ratings, as women with higher levels of intrasexual competition were more likely to enhance their appearance when viewing larger breast sizes (C and D) compared to smaller sizes. However, it should be cautioned that large breast sizes are not necessarily the most attractive size across all cultures, as average or slightly above the average sizes are considered most appealing in some non-Western societies [15,17,19,25]. These findings coupled with the evidence of difficulty in milk production and breastfeeding in women with large and very large breast sizes [20–23] suggests that largest breast sizes might not necessarily induce the highest competition and threat in women from every culture. Accordingly, in addition to the current findings among Hispanic women, the effect of breast size and shape on women's competition and threat inducement across different cultures should be investigated in the future research. Intrasexual competition levels in women were also associated with increase ratings of dieting and exercise and perceiving the woman as a sexual threat. This suggests that women who report higher levels of intrasexual competition are more likely to engage in enhancing behaviors such as dieting and exercising and perceive breasts in general as potentially sexually threatening.

The findings of the current paper extend the notion that women have psychological mechanisms that can detect and compete with other women [29] to features such as women's breast size and shape. Because men prioritize attractiveness and find specific breast features attractive, women may compete with other women to enhance their own features that are in line with men's evolved preferences. Since women with large non-ptotic are perceived to be associated with residual reproductive value [4,10], viewing women with these breast features may enhance intrasexual competitiveness in women. This may result in women exploiting these evolved preferences and competing with other women by increasing their own attractiveness through enhancing their appearance. Research has shown that women are likely to enhance their appearance [32,44], derogate competitive rivals [33], and guard their mates in the presence of women who are attractive [57–59]. Women also perceive other women with large firm breasts as a threat, and less likely to introduce them to their current partner, possibly to prevent desertion [4].

Further, this study shows that in addition to a general likelihood to enhance their appearance, women may also be more likely to wear revealing clothing and diet and exercise when viewing women with large non-ptotic breasts and intermediate distances. Women who show more cleavage have been shown to be judged more negatively [60]. This may indicate that they are viewed as rivals and competitors, and possibly sexually permissive [60,61]. This, in turn, may influence women to compete directly with those women to be able to self-promote their appearance to increase their mate value when competing for mates [52]. One explanation for why women may wear revealing clothing and diet and exercise when viewing women with attractive breasts features, is that they may be perceived as threatening their current relationships or current mating opportunities. Indeed, we did find that women considered women with large, low ptotic breasts with intermediate distances as a sexual threat, which would be in line with women considering other women with attractive features as a threat to their relationship. In support of previous research, women were more likely to diet and exercise when viewing breasts that were D-cup, low-ptotic, and intermediate distances. This highlights women's strategies in appearance enhancements such as lowering overall body fat and increasing their physical attractiveness through lowering BMI and waist to hip ratio, accentuating body curvaceousness [52]. More so than men, women are more willing to lose weight and diet [31], make social comparisons that lead to increasing diet pill usage [59], and increase dieting in the presence of attractive women [32]. This suggests that women calibrate their appearance enhancement when faced with attractive women in the environment to keep or maintain a current partner or compete with other women. These calibrations may include engaging in behaviors that increase traits that are attractive to men, such as lower BMI [62] and wearing revealing clothing [30].

The current study has a few limitations. First, we did not investigate whether women's appearance enhancement was associated with menstrual cycle status or ovulation probability, as research has indicated that conception status may increase enhancement strategies [30,63] and mate retention [58]. Investigating whether enhancement strategies increase across the menstrual cycle in the presence of women with variations in breast morphology is an avenue for future research. Second, we did not consider the role of relationship status, which has been shown to drive enhancement appearance and mate retention when exposed to an attractive woman [32]. Perhaps including relationship status and relationship satisfaction can further elucidate if women may alter their enhancement strategies when viewing women's breasts. Third, we only considered the role of women's breasts in influencing women's enhancement strategies. Research has shown that other features in addition to women's breasts, such as waist to hip ratios (WHR) and facial femininity [47] can play a role in how threatening women may appear. It is possible that using WHRs and facial femininity can increase women's strategies, as men also consider those traits highly attractive. Fourth, using a small sample of colleague students which is not representative of Hispanic culture is another limitation of the current research that needs to be address in the future research through collecting larger and more diverse Hispanic participants. Another limitation of

this study is the use of stimuli that are created using 3D models which might have resulted in lower ecological validity compared to photos of actual breasts. Yet, these models have enabled us to systematically manipulate the variables of interest while keeping other factors such as BMI, body shape, skin color, etc., constant across the stimuli. Moreover, using multiple stimuli asking for similar ratings might have resulted in participant fatigue and/or uncomfortableness. Future research might choose to consider between-subjects design both for the measured traits (breast size, ptosis, and cleavage) as well as the questions (likelihood of enhancing appearance, sexual threat, etc.). Additionally, a between-subjects design would eliminate any potential chance of participants' response adjustments across dependent variables. Finally, while the current study manipulated multiple breasts traits (i.e., breast size, cleft distance, and ptosis), a female model with a constant body size and BMI was used. Previous research has shown that breast size and BMI are correlated [64], and only during the final generation of ratings in multivariate selection analyses breast size become an important factor in attractiveness judgments by men [65]. This points to the limitation of the current research as well as importance of multivariate approaches in mate selection using multiple physical characteristic [65,66]. Evolution of attractive traits in humans through sexual selection is complicated and deserves a comprehensive approach, including extended phenotype theory, life history theory, and behavioral ecology [67,68].

## 5. Conclusions

The study showed that breast morphology that is indicative of residual reproductive value affected women's enhancement strategies independent of being primed with an intrasexual competition prime. Women were more likely to report enhancing their appearance, wear revealing clothing, diet and exercise, and rate the women they viewed as a threat if possessing large breasts that were non-ptotic and with intermediate intermammary distances. The findings suggests that women are attentive to breast morphology that are considered attractive to men and may engage in enhancement strategies to increase their own desirability and mate value.

**Supplementary Materials:** The following supporting information can be downloaded at: https://www.mdpi.com/article/10.3390/sexes4010008/s1. File S1: Summary of Mixed ANOVA tables.

**Author Contributions:** Conceptualization, R.G. and F.P.; methodology, R.G. and F.P.; formal analysis, R.G.; writing—original draft preparation, R.G.; writing—review and editing, R.G. and F.P. All authors have read and agreed to the published version of the manuscript.

**Funding:** This research received no external funding.

**Institutional Review Board Statement:** The study was conducted in accordance with the Declaration of Helsinki, and approved by the Institutional Review Board of Texas A&M International University.

**Informed Consent Statement:** Informed consent was obtained from all subjects involved in the study.

**Data Availability Statement:** Data will be made available upon request to corresponding author.

**Conflicts of Interest:** The authors declare no conflict of interest.

## Appendix A

Scenario 1

Imagine you and your partner are at a restaurant and you continuously notice his eyes following a waitress around the restaurant. As the meal goes on, you notice he is paying less attention to you, and more attention to her. You continue to sit there and not say anything about what you notice. After a while, he makes a comment about how attractive he finds her and you think you hear him quietly say to himself he wishes he had met her before you, because he would love to be with her.

Scenario 2

Imagine you have been dating your partner for four years. At every party, you notice this one girl who you do not know is always trying to get close to him and take him away from the group. As the school year continues, you notice at parties he is paying less and less attention to you, and more attention to this girl. You realize they are directly flirting and they are not trying to hide it. You overhear two people talking about how they cannot believe that after the last party, your partner went home with this girl.

Scenario 3

You see your partner's phone on a kitchen table as you walk away; it rings to indicate a text. You go to bring his phone to him in the living room but notice there is a girl's name and a lot of hearts. Another text comes in and you see that she is calling him "sexy". You hand him his phone, he looks at the screen, turns it off, and doesn't say anything.

Scenario 4

You have been in a relationship with your partner for six months but not all your friends are aware of it yet, as you are in a different group of friends. One night you go out with a group of your friends, and your partner is not there. After a few drinks, this woman that came with one of your friends starts telling a story about this new boy she is seeing. She is going on about how hot he is, very sexy, and tall. When another friend asks her what the boy's name is, she says it, and you realize it is your partner.

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
