# Peer review of "Intrasexual Competition in Women’s Likelihood of Self-Enhancement and Perceptions of Breast Morphology: A Hispanic Sample"

_sexes, doi:10.3390/sexes4010008_

Round 1
Reviewer 1 Report
Review of “Intrasexual competition in women’s likelihood of self-enhancement and perceptions of breast morphology: A Hispanic sample”
Manuscript number: sexes-2078476
In this paper, the authors extend evolutionary research on women’s intrasexual competition by focusing on the residual reproductive information that can be gleaned from breast morphology and how this information may be used in intrasexual competition. Specifically, the researchers aimed to determine if variations in breast size, ptosis, and intermammary distance influenced women’s likelihood of enhancing their appearance, wearing revealing clothing, dieting and exercising, and viewing the breasts as a sexual threat. The authors found evidence that women were more likely to report an increased likelihood of enhancing their appearance, wearing revealing clothing, dieting and exercising, and perceiving the breasts as a sexual threat when they were shown pictures of larger breasts with low ptosis and intermediate distances. The authors also investigated other potential influences such as partner threat and individual differences in intrasexual competition but found that these factors had little to no influence on the majority of their effects.
In all, the manuscript provides a logical extension of the research on women’s intrasexual competition that exists in the literature. Additionally, the authors also note that they went above and beyond to ensure that their sample included a traditionally understudied population in evolutionary research. However, even with these strengths, I do have some outstanding concerns about this manuscript that I feel the authors would need to address in order for this manuscript to be suitable for publication. I am not sure if the authors will be able to fully address some of these concerns without incorporating another study, so I, therefore, recommend a major revision.
Below I provide details about my concerns, enumerated for ease of communication during the review process.
-
While the researchers provide some rationale that breast size may be an indicator of fertility, residual reproductive value, and reproductive success, this connection has already been established preliminarily in the literature (Havlíček et al. 2017). But, what this and the previous literature has overlooked is that there does not appear to be a connection between breast size and the ability to lactate/ milk production (Żelaźniewicz and Pawłowski 2019). In fact, women with larger breasts often times report that they have a harder time breastfeeding compared to their counterparts with smaller breasts (Katz et al. 2010; Massov 2015; Chang et al. 2020). These effects are often also found in studies looking at overweight women, and body image issues, and often rely on self-reports, but this is information that IBCLCs and other lactational support professionals commonly use and report to women who frequent their clinics. This begets an interesting point that the authors' results seem to support (at least from my perspective), the largest breast size may not be the most desirable/competition-inducing simply because women who had this size (ancestrally) would have had a harder time breastfeeding and ensuring that children born to them survived and passed on that same preference. However, with the modern technological invention of formula, male preferences may be experiencing runaway selection now that infant survival isn’t as tightly coupled with breastfeeding success. While I do not want to imply that the authors endorse this perspective or change their hypotheses, I do think the fact that this relationship between breast size, breastfeeding, and infant survival exists needs to be discussed in the discussion to help contextualize the results. Additionally, it is mentioned later that breast morphology changes associated with pregnancy/ parturition are deemed as unattractive in humans, so it may be important to point out that humans appear to be the only primate that has this preference (since male chimpanzees and baboons appear to prefer non-nulliparous mates who have already shown that they can successfully care for dependent offspring).
-
I found it interesting that the authors talk about different dimensions of what about breasts is found attractive, but do not talk about the non-cosmetic approach that some women (especially young women) use to enhance their own intermammary distance: using push-up bras. The use of these bras provides a modern technological advantage but, as with make-up usage, seems like a plausible invention that women could use to help them make themselves more attractive to others and appear more competitive.
-
While I appreciated the connection to the lumbar curvature research, it did not seem to fit with the flow of the introduction. I think its inclusion is important and helps to set up the rationale of the current study, but I think that the authors need to improve the writing flow around this point so that it feels more cohesive with the rest of the introduction.
-
One main concern that I have is that the number of statistical tests run seems quite high for such a small sample size. I understand that the authors performed corrections for multiple tests, but that does not change the fact that their analyses may be underpowered to begin with. Did the authors run a priori power analyses or post hoc sensitivity analyses to ensure that they are properly powered? If not, I am inclined to request that the authors include an additional study to ensure that these effects hold up when the analyses are properly powered.
-
While the authors have made a point of indicating that they focused on a traditionally understudied population, I feel like this point needs to be expanded upon in order for researchers who are not familiar with evolutionary psychology (and its pitfalls) to understand this significance. Was this a strategic choice to address concerns about the overrepresentation of White participants in what we think of as “universal” psychological features? Due to the nature of the college population? Either is a fine rationalization, but how the authors have included this information currently feels like a throwaway sentence that does not fully explain the importance of this decision.
-
More information is needed on the partner threat primes. When reading the measures section, it first appeared as if the participants only saw one prime. But, when I got to the procedures section, it was clear that the participants in the threat condition saw all four primes. Why did participants see all four conditions? This seems like overkill if the researchers truly believed that the threat prime should evoke more competitive responses and this may have washed out any potential effects because participants were bombarded with this. Are there substantive differences in the four primes? Do they build on one another? I am not familiar with these primes and would strongly encourage the authors to provide more details on them (perhaps providing an example) instead of directing readers to a different paper that may be behind a paywall for some readers.
-
The fact that participants were shown 36 pictures of breasts and asked to rate them has me concerned that a lot of these effects may be the result of demand characteristics. It is commendable to investigate each of these potential influences, but to combine them in this fashion really makes me think that the purpose of the study was not sufficiently kept from participants and makes me worry that these results are not accurate. I also worry about participant fatigue and uncomfortability when looking at these images, since participants were asked to make the same ratings after each image (if I read the paper correctly). While the authors cannot undo this in the current form, I do think there needs to be a stronger rationale as to why they chose so many images and had the same ratings after each. In addition to this, I feel that additional studies are needed to show these effects hold up when the authors are not presented with such an overwhelming number of photos
-
I feel like more information is needed to understand how the dependent variables were presented to participants and if there was any cover story told to participants to prevent them from guessing the purpose of the study and responding to demand characteristics. How it is currently described, it really seems like participants could have easily gleaned that the researchers were interested in how women responded to these images and tried to “help” by making themselves respond more competitively across DVs.
-
As expressed earlier, I am worried that this investigation is not properly powered for a single 2x4x3x3 mixed ANOVA let alone 4 mixed ANOVAs and an LME. What was the rationale for using such complicated tests instead of investigating one feature at a time between conditions? Was there a theoretical reason to expect that participants would need to see these features altogether instead of separately? I feel like this is my biggest concern for this study since the use of corrections only corrects for multiple tests but not the possibility that these analyses are underpowered. The results are easy enough to follow, the interactions appear on the surface to be easily interpretable (though I must comment that I am not sure that the higher level interactions were investigated given that they are not mentioned here even though some of the results appear to hint that there could have been interactions - this could easily be rectified by incorporating a table listing all of the main effects and interactions regardless of whether they indicated significance).
-
The authors concluded that the partner threat prime did not affect ratings but it feels like a simpler explanation is that, due to seeing four primes regarding partner threat condition, any effect that may have been there because the primes could be too overwhelming or lack social validity. It is hard to make any suggestions on this as a reviewer since the primes were not given in the manuscript and the authors did not present a rationale for including all 4 in that condition.
-
The authors mention that their results support previous research because women who saw D-cup, low ptotic, and intermediate distance breasts were most likely to report that they would diet and exercise, but this seems odd that research suggests that women lose weight in the breasts first when engaging in dieting and exercise. I think the authors could strengthen this explanation by highlighting how diet and exercise may increase their attractiveness in other mating-related traits (such as lowering WHR, lowering overall body fat, etc.) so that it is clear the authors are not suggesting that diet/ exercise will make women competitive within this same domain.
-
I feel like the authors did not provide a good-faith examination of the limitations of their study. For example, they have a fairly small sample size, use an extremely (and potentially unnecessarily) complex design, only have college-attending participants, and have stimuli that may lack ecological/ social validity, but instead of mentioning these limitations focus on what they did not measure. I would suggest that the authors focus on the former in the limitations, rather than the latter because there will always be something that isn’t measured but these current limitations need to be addressed in order for the current study to have more believability.
On the whole, I enjoyed this manuscript but do think there are some serious revisions (and the potential inclusion of additional studies) that need to be made before this manuscript is ready for publication. I hope the authors find these comments with the constructiveness with which they are intended.
As per my policy, I sign all of my reviews. The author should feel free to contact me if they have any questions or points of clarification regarding our review.
Reviewer 2 Report
The current study tested how aspects of breast morphology (size, shape and cleft distance) in other women determines women’s appearance enhancing behaviours. The results showed women’s appearance enhancement behaviours were greater following exposure to stimuli depicting fuller, firmer breasts with more equidistant alignment. This suggests a role for intra-sexual selection in shaping appearance modification in women.
The study is interesting, well conducted and correctly analysed. I just have some very minor comments for the authors.
Introduction
1. How sexual selection has shaped the evolution of attractive traits is complicated in humans, who may embellish their appearance via cultural modifications (Dixson, 2019; Luoto, 2019). Davis and Arnocky (2020) suggest cosmetic surgery to alter breast morphology could serve to enhance attractiveness to men and function during intra-sexual competition between women. As breast size, shape, and firmness change with age and parity, young nulliparous women may opt to cosmetically alter breast size to enhance attractiveness or intra-sexual competitiveness depending on the distribution of breast size among their peers (Dixson, 2022).
REFERENCES
Dixson, B. J. (2019). Sexual selection and extended phenotypes in humans. Adaptive Human Behavior and Physiology, 5, 103-107.
Dixson, B. J. (2022). Sexual selection and the evolution of human appearance enhancements. Archives of Sexual Behavior, 51, 49-55.
Luoto, S. (2019). An updated theoretical framework for human sexual selection: from ecology, genetics, and life history to extended phenotypes. Adaptive Human Behavior and Physiology, 5, 48-102.
2. There are some additional papers using eye-tracking showing that breast morphology and lower body adiposity captures men’s attention, attractiveness judgments for specific sizes vary (Dixson et al., 2010; 2011; Gervais et al., 2013). These should be mentioned in the introduction.
REFERENCES
Dixson, B. J., Grimshaw, G. M., Linklater, W. L., & Dixson, A. F. 2010. Watching the hourglass: Eye-tracking reveals men’s appreciation of the female form. Human Nature, 21, 355-370.
Dixson, B. J., Grimshaw, G. M., Linklater, W. L., & Dixson, A. F. (2011). Eye-tracking of men’s preferences for waist-to-hip ratio and breast size of women. Archives of Sexual Behavior, 40, 43-50.
Gervais, S. J., Holland, A. M., & Dodd, M. D. (2013). My eyes are up here: The nature of the objectifying gaze toward women. Sex Roles, 69, 557-570.
3. The authors cite Havilcek’s et al (2017) cross-cultural study of male preferences for female morphology. This was a replication of another cross-cultural study found that men from remote resource scarce communities in the highlands of Papua New Guinea had stronger preferences for larger breasts than men from Samoa and New Zealand (Dixson et al., 2011). This study should be referenced along with the replication.
REFERENCE
Dixson, B. J., Vasey, P. L., Sagata, K., Sibanda, N., Linklater, W. L., & Dixson, A. F. (2011). Men’s preferences for women’s breast morphology in New Zealand, Samoa, and Papua New Guinea. Archives of sexual behavior, 40, 1271-1279.
Discussion
The current study manipulated breast size, cleft distance and ptosis on computer-generated female torsos. While this is a multivariate manipulation, there is still the possibility that effects were influenced by trait interactions rather than the discrete manipulations. One example is applying manipulations to breast size while holding BMI constant, where in reality BMI and breast size are likely positively correlated. Brooks et al (2015) used multivariate selection analyses over eight generations of attractiveness ratings showed that when allowed to evolve freely under sexual selection, men’s preferences were strongest for slim bodies with narrow waists and only during the final generation of ratings did breast size become an important factor in attractiveness judgments. This study should be included in the discussion to highlight a limitation of stimuli used in the current experiment.
REFERENCE
Brooks R. C., Jordan, A.L., Shelly, J., Dixson B. J. 2015. The multivariate evolution of female body shape in an artificial digital ecosystem. Evolution and Human Behavior, 36, 351-358.
Round 2
Reviewer 1 Report
Review of “Intrasexual competition in women’s likelihood of self-enhancement and perceptions of breast morphology: A Hispanic sample” revision
Manuscript number: sexes-2078476 (revision)
In this paper, the authors extend evolutionary research on women’s intrasexual competition by focusing on the residual reproductive information that can be gleaned from breast morphology and how this information may be used in intrasexual competition. Specifically, the researchers aimed to determine if variations in breast size, ptosis, and intermammary distance influenced women’s likelihood of enhancing their appearance, wearing revealing clothing, dieting and exercising, and viewing the breasts as a sexual threat. The authors found evidence that women were more likely to report an increased likelihood of enhancing their appearance, wearing revealing clothing, dieting and exercising, and perceiving the breasts as a sexual threat when they were shown pictures of larger breasts with low ptosis and intermediate distances. The authors also investigated other potential influences, such as partner threat and individual differences in intrasexual competition. Still, they found that these factors had little to no influence on most of their effects.
I appreciate the chance to re-review the manuscript in light of the changes that the authors have made. They have adequately incorporated the majority of the suggestions that the other reviewer and I have made, and I feel that their manuscript is greatly improved. However, there are a few outstanding issues that need to be addressed before the manuscript can be published. I have quoted the text from the response to the reviewers here to help with the reviewing process. My original comment is in normal text, the author’s response is bolded, and my new response is in italics.
-
While the researchers provide some rationale that breast size may be an indicator of fertility, residual reproductive value, and reproductive success, this connection has already been established preliminarily in the literature (Havlíček et al. 2017). But, what this and the previous literature has overlooked is that there does not appear to be a connection between breast size and the ability to lactate/ milk production (Żelaźniewicz and Pawłowski 2019). In fact, women with larger breasts often times report that they have a harder time breastfeeding compared to their counterparts with smaller breasts (Katz et al. 2010; Massov 2015; Chang et al. 2020). These effects are often also found in studies looking at overweight women, and body image issues, and often rely on self-reports, but this is information that IBCLCs and other lactational support professionals commonly use and report to women who frequent their clinics. This begets an interesting point that the authors' results seem to support (at least from my perspective), the largest breast size may not be the most desirable/competition-inducing simply because women who had this size (ancestrally) would have had a harder time breastfeeding and ensuring that children born to them survived and passed on that same preference. However, with the modern technological invention of formula, male preferences may be experiencing runaway selection now that infant survival isn’t as tightly coupled with breastfeeding success. While I do not want to imply that the authors endorse this perspective or change their hypotheses, I do think the fact that this relationship between breast size, breastfeeding, and infant survival exists needs to be discussed in the discussion to help contextualize the results. Additionally, it is mentioned later that breast morphology changes associated with pregnancy/ parturition are deemed as unattractive in humans, so it may be important to point out that humans appear to be the only primate that has this preference (since male chimpanzees and baboons appear to prefer nonnulliparous mates who have already shown that they can successfully care for dependent offspring).
Response: Thank you for this fair and interesting point. We have added the mentioned references in the Introduction (i.e., Żelaźniewicz and Pawłowski 2019; Katz et al. 2010; Massov 2015; Change 2020) pointing to this fact that women with obesity and large breasts might have difficulty in breastfeeding. While we appreciate reviewer’s suggestion about the largest breast size may not be the most desirable/competition-inducing, however, our results showed D-cup breasts on the other hand were considered more sexual threat “D-cup breasts (M = 4.37, SE = .12) were rated as being sexually threatening compared to C- (M = 3.97, SE = .11), B- (M = 3.17, SE = .09), and A-cup breasts (M = 2.55, SE =.11).”, as well as resulting in more appearance enhancement “For appearance enhancement, women’s intrasexual competitiveness moderated the likelihood of enhancing appearance when viewing D-cup (b = .05, SE = .004, p < .001), C-cup (b =.03, SE = .004, p < .001), but not B-cup breasts (b = .006, SE = .004, p = .12”. We appreciate reviewer’s interesting point about this association between large breast sizes and lower desirability/competition-inducing, however, we refrain from discussing in that direction as our results suggests otherwise.
Thank you for making some of these changes. However, I must point out that failure to include relevant literature in an introduction section that provides a plausible alternative hypothesis simply because the results do not support that interpretation is not a valid response. While your results suggest one interpretation, it is not the only interpretation in the literature (see (Kościński et al. 2020; Dixson et al. 2011; Dixson et al. 2015; Gründl et al. 2009; Swami and Tovée 2013; Zelazniewicz and Pawlowski 2011) for research suggesting that average or slight above average sized breasts are the most attractive). In the US, DD is considered to be the average cup size, but this is not true across the world. In Mexico and Japan, for example, A is considered to be the average cup size (https://worldpopulationreview.com/country-rankings/breast-size-by-country). The justification for not including these studies in the introduction because your results do not support them is not sufficient because the point of research is to survey the vast amount of other evidence with novel data before providing a new perspective. Again, I am not asking you to change your hypotheses or say that this new research should outweigh your current rationale, but to not include it because your results don’t support it is not sufficient.
-
One main concern that I have is that the number of statistical tests run seems quite high for such a small sample size. I understand that the authors performed corrections for multiple tests, but that does not change the fact that their analyses may be underpowered to begin with. Did the authors run a priori power analyses or post hoc sensitivity analyses to ensure that they are properly powered? If not, I am inclined to request that the authors include an additional study to ensure that these effects hold up when the analyses are properly powered.
Response: We understand the reviewer’s concern, and we did run an a priori power analysis. We ran a power analysis for a mixed anova and the power analysis revealed that a sample size of 182 participants to detect a 80% power using a small to moderate effect size of f = .15 (a = .05) would be a sufficient sample size. On page 3, line 134-135, we included the following in the participants section of the methodology, “An a-priori power analysis using G*Power (f = .15, a = .05) revealed that a sample size of 182 participants would be an adequate sample to detect a small to medium effect size”
Unfortunately, this does not address my concern. You have provided power analyses for a single mixed ANOVA, not power analyses for the four that you run. Your experiment-wise alpha level (https://www.real-statistics.com/one-way-analysis-of-variance-anova/experiment-wise-error-rate/) is much higher than 0.05 as a result. To correct this, please adjust your alpha level so that the experiment-wide alpha is within the correct range. Given the complicated nature of the design, it seems imperative to me that you ensure you are properly powered with such a small sample when multiple tests are run.
-
The fact that participants were shown 36 pictures of breasts and asked to rate them has me concerned that a lot of these effects may be the result of demand characteristics. It is commendable to investigate each of these potential influences, but to combine them in this fashion really makes me think that the purpose of the study was not sufficiently kept from participants and makes me worry that these results are not accurate. I also worry about participant fatigue and uncomfortability when looking at these images, since participants were asked to make the same ratings after each image (if I read the paper correctly). While the authors cannot undo this in the current form, I do think there needs to be a stronger rationale as to why they chose so many images and had the same ratings after each. In addition to this, I feel that additional studies are needed to show these effects hold up when the authors are not presented with such an overwhelming number of photos.
Response: We followed the same approach used by Garza et al. (2020, 2022) and Pazhoohi et al. (2020), where 36 images were used for the within factor variables. We included in the procedure on page 4, lines 185-189, “We followed the same approach from previous studies [3, 4, 10], where they viewed the 36 breast images, and indicated their likelihood to enhance their appearance, wear revealing clothing, diet and exercise, and the likelihood of the woman viewed being a sexual threat.” We have confidence that the results are accurate, as they are similar to the findings shown in Garza et al. (2022), where women typically find larger breasts threatening, and in this study we expand on how women’s breasts may impact women’s intrasexual competitive tactics. To add, we did not find any strong evidence that the experimental prime was driving those perceptions, instead focusing on the dispositional traits of intrasexual competition.
While I can understand that there is precedence for this methodology in the literature (and the previous studies both authors have conducted), this does not address my concern. There is still the possibility (and a very likely one in my opinion) that some of the responses are related to demand characteristics, participant fatigue, and uncomfortableness. Please discuss these as limitations of your study. The simplest future directions could follow from this, where other authors assess the extent to which these results are stable when only shown a single picture for each DV of interest.
-
I feel like more information is needed to understand how the dependent variables were presented to participants and if there was any cover story told to participants to prevent them from guessing the purpose of the study and responding to demand characteristics. How it is currently described, it really seems like participants could have easily gleaned that the researchers were interested in how women responded to these images and tried to “help” by making themselves respond more competitively across DVs.
Response: We would like to clarify that the dependent variables were presented on the same presentation page as the breasts. The breasts were randomly presented, and the three dependent variables were shown at the bottom of the screen. This approach is similar to studies aforementioned in the reviewer’s previous comment. In reference to the reviewer’s latter concern, it is unclear if the reviewer is referring to the manipulation as “helping” or the actual dependent variables. On page 4, lines 189-190, we included the following, “The ratings were presented at the bottom of the same page of the breast image presentation.”
Unfortunately, this does not address my concern about demand characteristics. I understand that this is a part of the design that can not be changed, but this is a limitation that needs to be addressed with this study.
-
As expressed earlier, I am worried that this investigation is not properly powered for a single 2x4x3x3 mixed ANOVA let alone 4 mixed ANOVAs and an LME. What was the rationale for using such complicated tests instead of investigating one feature at a time between conditions? Was there a theoretical reason to expect that participants would need to see these features altogether instead of separately? I feel like this is my biggest concern for this study since the use of corrections only corrects for multiple tests but not the possibility that these analyses are underpowered. The results are easy enough to follow, the interactions appear on the surface to be easily interpretable (though I must comment that I am not sure that the higher level interactions were investigated given that they are not mentioned here even though some of the results appear to hint that there could have been interactions - this could easily be rectified by incorporating a table listing all of the main effects and interactions regardless of whether they indicated significance).
Response: The rationale for using multiple features altogether (within), rather than between, is because women’s features vary by multiple physical dimensions. We followed with what has been done in the literature, and in research on breasts perceptions, it is uncommon that women’s breasts are shown separately, rather they are always combined with other features (i.e., ptosis, waist to hip ratios, nipple placement, to name a few). Below, we have included a list of studies where breast features are combined and not isolated. We also followed through with the reviewer’s suggestions and included the anova tables for the analysis.
Again, while I understand that there is precedence for this in the literature, this response does not address my concern. This is a limitation of the study that needs to be addressed. I also am not sure if I missed it or not, but there was not a table included in the new version of the manuscript that I have access to.
Again, I enjoyed this manuscript but do think these points needs to be seriously addressed before the manuscript can be published. If I, someone who agrees with the points, conclusions, and rationale made in the manuscript, still has concerns about these issues I would assume that others will as well (especially those who do not agree with the theoretical perspective and/or conclusions). I hope the authors find these comments with the constructiveness with which they are intended.
